# Eco-Geographical, Morphological and Molecular Characterization of a Collection of the Perennial Endemic Species *Medicago tunetana* (Murb.) A.W. Hill (Fabaceae) from Tunisia

**DOI:** 10.3390/plants10091923

**Published:** 2021-09-15

**Authors:** Yosr Ferchichi, Anis Sakhraoui, Hela Belhaj Ltaeif, Yosr Ben Mhara, Mohamed Elimem, M’barek Ben Naceur, Zeineb Ghrabi-Gammar, Slim Rouz

**Affiliations:** 1Laboratory of Agriculture Production Systems and Sustainable Development (LR03AGR02), Department of Agricultural Production, Higher School of Agriculture of Mograne, University of Carthage, Mograne-Zaghouan 1121, Tunisia; anis.sakhraoui@esakef.u-jendouba.tn (A.S.); belhajhela1@gmail.com (H.B.L.); yosrmhara13@gmail.com (Y.B.M.); mohammed.elimem123@gmail.com (M.E.); slim.rouz@esamg.u-carthage.tn (S.R.); 2National Institute of Agronomy of Tunis, University of Carthage, Tunis 1082, Tunisia; zghrabi@yahoo.fr; 3Higher School of Agriculture of Kef, University of Jendouba, Le Kef 7119, Tunisia; 4Departamento de Biología Vegetal y Ecología, Universidad de Sevilla, Apartado 1095, 41080 Sevilla, Spain; 5National Gene Bank of Tunisia, Boulevard Leader Yasser Arafat Z. I. Charguia 1, Tunis 1080, Tunisia; nour3alanou@yahoo.com; 6Laboratoire de Recherche Biogéographie, Climatologie Appliquée et Dynamiques Environnementales (Bi CADE 18ES13), Faculté des Lettres des Arts et des Humanités de Manouba, Campus Universitaire de la Manouba, Université de la Manouba, Manouba 2010, Tunisia

**Keywords:** characterization, conservation programs, genetic diversity, SSRs markers, polymorphism

## Abstract

In order to characterize and conserve the endemic pastoral species *Medicago tunetana*, many prospecting missions were carried out in mountainous regions of the Tunisian ridge. Twenty-seven eco-geographical and morphological traits were studied for six *M. tunetana* accessions and followed by molecular analysis using seven Simple Sequence Repeat (SSR). Only five markers were polymorphic and reproductible in the six *M. tunetana* populations. A total of 54 alleles were observed with an average of 10.8 bands/primer/genotype. Mean Polymorphism Information Content (PIC), Nei gene diversity (h) Shannon’s information index (I) indicated the high level of polymorphism. The generated dendrogram with hierarchical UPGMA cluster analysis grouped accessions into two main groups with various degree of subclustring. All the studied accessions shared 57% of genetic similarity. Analysis of variance showed high significant difference between morphological traits among *M. tunetana* populations where MT3 from Kesra showed different morphological patterns regarding leaf, pod and seeds traits. Canonical correspondence analysis (CCA) showed two principal groups of *M. tunetana* populations based on potassium, total and active lime contents in soil. Our results suggest that SSR markers developed in *M. truncatula* could be a valuable tool to detect polymorphism in *M. tunetana*. Furthermore, the studied morphological markers showed a large genetic diversity among *M. tunetana* populations. This approach may be applicable for the analysis of intra specific variability in *M. tunetana* accessions. Our study could help in the implementation of an effective and integrated conservation programs of perennial endemic *Medicago*.

## 1. Introduction

Tunisia has a large and wealthy forage and pasture biodiversity. According to Abdelguerfi and Abdelguerfi-Laouar [1], there are more than 960 pasture legumes species, 336 of which are Mediterranean endemic. However, native and endemic Tunisian pasture plants are endangered due to many factors among which overgrazing [2] and natural habitat loss into introduction of intensive farming and which are the most important ones [3,4]. Thereby, conservation and valorisation of pasture and forage genetic resources have become key tasks, especially in Tunisian Dorsal regions where pastoralism is an important genetic loss drivers. However, pasture genetic resources in Tunisia are still less developed, despite the fact that they can enrich the Gene Bank and include many varieties in breeding [5].

Lucerne (*Medicago sativa* L.) is also known as queen forage alfalfa [6] thanks to its ecological attributes by avoiding erosion phenomena [3], and its wide agronomic assets by way of its protein and nitrogenous matter contents [7,8]. Perennial species of *Medicago* genus have an essential role in the economic sustainability of crop-livestock systems [9]. Nevertheless, alfalfa productivity is limited by some abiotic stress such as salinity [10,11]. *Medicago tunetana* (Murb.) A.W. Hill is a perennial C3 species native to Algeria and Tunisia thriving in calcareous mountains of Western North and Midwest regions of Tunisia [12]. Tunisian alfalfa has many interested eco-physiological assets such as drought tolerance, winter hardiness and calcareous soil tolerance; this genetic resource has been unstudied so far and threatened to disappear. Furthermore, the efficient management of this pasture plant species in its native regions of the Tunisian ridge may contribute in the agro-pastoral systems development. The plant breeding for the fodder crops like alfalfa is of a high importance especially for the farming system sustainability [9]. For an optimal valorisation, it is necessary to conserve, evaluate and estimate the genetic and morphological variation of *M. tunetana* and to determine the relationship within this endemic species and cultivated alfalfa. In fact, there is an ambiguity in *M. tunetana* taxonomy; it is considered as a subspecies of *M. sativa* [13,14,15]. However, and according to other authors, *M. tunetana* is classified as an independent species [16,17,18].

Molecular characterization has become the most efficient and reliable tool for studies of the genetic diversity between populations and species as well thanks to its environment independency [19,20]. In fact, it is not influenced by the stage of plant development facilitating genetic resources management within Gene Bank [21]. Molecular markers are complementary to morphological and biochemical markers [22]. Several studies have been realized on the genetic diversity of *Medicago* genotypes using different markers such as AFLP [23,24], RAPD [25,26], RFLP [27,28] and SSR [29,30,31,32]. The microsatellites as polymorphic DNA markers have been widely employed in molecular studies thanks to their facility and efficiency for population genetic analysis [33,34,35] while, using SSR markers for genetic variability within perennial *Medicago* genotypes had not been very developed. Diwan et al. [36] are the first authors to demonstrate the use of SSRs to characterize genetic diversity and to analyse the genetic relationships among *Medicago* genotypes as well. Cholastova and Knotova [30] indicated that the genetic diversity estimation of *Medicago* genotypes could be obtained even with only three SSR markers. Furthermore, 107 SSRs identified in the EST database of *Medicago truncatula* Gaaertn. was mapped also in *M. sativa* [37]. 

In order to study the *M. tunetana* collection, an ecological and edaphic characterization of different prospected sites was realized and followed by morphological and molecular characterization for seven accessions of *M. tunetana* and one variety of *M. sativa* used as a reference. For the molecular characterization, seven microsatellites markers were used in order to estimate the genetic diversity among *M. tunetana* local genotypes and to analyse the genetic relationship between *M. tunetana* and *M. sativa*.

## 2. Results

### 2.1. Ecology of M. tunetana Sites

*M. tunetana* was recorded in several new sites in the Tunisian dorsal region that were not mentioned in relevant flora (Figure 1). These sites are characterized by a large bioclimatic zone differences ranging from a sub-humid with cool winter at Sekiet Sidi Youssef to a medium semi-arid at Kasserine sites (Table 1). *M. tunetana* generally grows at the canopy of *Pinus halepensis* Mill. plantations characterized by a high level of organic matter (OM) in the soil that can reach more than 6% like for MT3 and MT4 soils in Kesra and Sekiet Sidi Youssef with an average of 8.6 and 6.72% of OM respectively. The statistical analysis concerning edaphic parameters showed a significant difference at *p* < 0.05. In fact, two types of soil are registered: a heavy soil for MT2 and MT7 in Makthar and El Ayoun with an average of 37.2 and 37.88% of clay content respectively and a sandy soil for all other prospected sites. The values of soil pH in this study ranged from 7.2 to 8.36 indicating an alkaline nature of soil for all *M. tunetana* sites (Table 1). On the other hand, the total limestone and active limestone content demonstrated that all sites are a calcareous soil especially for the MT2 soil from Makthar with a rate of 30.7% of active limestone in the soil. The analysis of potassium (K) content in soil was measured and the statistical analysis showed a significant K content differences which is ranging between 38 ppm for MT1 from Bargou to 297 ppm for MT3 from Kesra.

### 2.2. Morphological Variation

The difference between seven *M. tunetana* accessions is statistically significant (*p* < 0.05) for eleven morphological traits as shown in Table 2. The highest morphological traits values relating to leaves, pods and seeds are obtained in MT3 from Kesra with an average of 17.06 mm of LL, 8.91 mm of PoW and 3.01 of SL followed by MT4 from Sekiet Sidi Youssef and MT1 from Bargou.

### 2.3. Canonical Correspondence Analysis

In order to estimate the relative importance of each eco-geographical trait in *M. tunetana* accessions distribution, the canonical correspondence analysis (CCA) was determined. Figure 2 demonstrated that K content and the total and active lime contents (CT, CA) were the most effective factors for the accession’s distribution. Two principal clusters are obtained; the first cluster which is characterized by a high K content level included the accession MT2 from Makthar.The second cluster includes MT3, MT4 and MT5 from Kesra, Sekiet Sidi Youssef and Thala respectively based on the total and active lime content in the soil. Moreover, CCA axis 1 is positively correlated with K, pH and Organic matter (OM%) content and negatively correlated with annual precipitation, longitude, latitude and Sand content (S%).

### 2.4. Genetic Diversity Analysis

The extent of polymorphism in seven accessions of *M. tunetana* (MT1-7) and a one variety of *M. sativa* (MS1) were analysed in 20 individuals from each genotype. Among the seven SSR markers used in this study, only five (MTIC82, MTIC338, MTIC343, FMT13 and B14B03) were polymorphic with *M. tunetana* genotypes (Table 3). Allele doses from five microsatellite loci were counted; a total of 54 alleles were detected in the analysed accessions with an average of 10.8 bands per primer and per genotype.

The genetic diversity of *M. tunetana* was demonstrated using the observednumber of alleles(na), effective alleles (ne), Shannon’s Information index (I) and Nei’s gene diversity (h). In fact, a high level of genetic diversity was recorded with an average of effective number of alleles (ne) that ranged between 1.2677 and 1.9231 (Table 4). MT4 genotype from Sekiet Sidi Youssef and MT6 from Thala showed the highest level of effective number of alleles (ne) and Shannon’s Information index (I) and Nei’s gene diversity (h) with an average of 1.9231, 0.48 and 0.6730 respectively. 

Genetic distance between pairs of genotypes was estimated using Matrix of Nei’s Distance. The largest distance was observed between MT2 and MT5 (0.8000) from Makthar and Thala respectively (Table 5) while the shortest distances were obtained between MT3 and MT7 (0.1278) collected from Kesra and El Ayoun respectively.

The dendrogram (Figure 3) presents the genetic relationship between the six *M. tunetana* accessions (MT1, MT3-7) and the reference variety of *M. sativa* (MS1) based on the genetic similarity coefficients obtained with UPGMA. Obtained results accord to the dendrogram and demonstratethat 57% of genetic similarity between *M. tunetana* genotypes have defined two clusters, Figure 3.
**Group 1 (G1):** Formed by two subgroups at 61% of genetic similarity with other accessions.−**Subgroup 1:** Formed by MT1 from Bargou and MS1, the variety of *M. sativa*.−**Subgroup 2:** Included four accessions of *M. tunetana*; MT3, MT5, MT4 and MT7 from Kesra, Thala, Sekiet Sidi Youssef and El Ayoun respectively.
**Group 2 (G2):** Formed by MT6 from Thala with a 57% of similarity to other accessions of *M. tunetana* and the reference variety of *M. sativa*.

## 3. Discussion 

*M. tunetana* had been found in three new prospected sites of the high mountains of Tunisian Dorsal namely Bargou (MT1), El Ayoun (MT7) and Thala (MT5) (Figure 1). These sites were not mentioned in Tunisian flora [43]. It must be noted that annual precipitation is ranging between 522 mm for Bargou and 299–327 mm for the others dorsal sites for the last twenty-seven year period (Table 1). For annual average temperature, Makthar presented the lowest value of 19 °C, while the highest value was recorded by the Southern dorsal regions with an average of 20.5 °C [4]. *M. tunetana* is native to Western North Tunisia and Algeria. It has many important ecological and agronomical interests including abiotic stress tolerance (cold tolerance, rhizome production and calcareous soil tolerance). These results are consistent with those obtained by El Makki-Ben Brahim et al. [4] who mentioned that the dorsal regions are characterized by a calcareous soil and those of Ferchichi and Rouz [43] who demonstrated that *M. tunetana* genotypes are tolerant to limestone excess in soil. Conservation and management of this rare pastoral genetic resource is an essential preoccupation in order to contribute to the grassland amelioration of the mountains of Western North and Western Centre regions of Tunisia.

In this study, we investigated the genetic diversity of seven wild accessions of *M. tunetana* and local variety of *M. sativa* as reference using seven SSR markers. We obtained 54 different alleles for a sample of 20 plants per accession (Table 3). While Li et al. [20] have obtained just 22 different alleles in *M. sativa* subsp. *falcata* populations using the same number of SSR markers for a sample size of 25 plants. Whereas Falahati et al. [29] obtained 68 alleles using eight SSR markers for a sample size of only 10 plants for each accession. Therefore, Julier et al. [44] noted that a sample size of 40 plants is more moderate for tetraploid alfalfa genetic diversity. In fact, these results are consistent with those of Andru [45] who reported that the number of observed alleles is highly dependent on the size of the studied samples. A sample size lower than ten plants gives rise to the loss of even non-rare alleles [44]. The studied *M. tunetana* genotypes show a large genetic variation with two different groups and two subgroups for only seven accessions (Figure 3) which confirms the hypothesis of tetraploidy for this species. The high number of alleles per locus may be explained by the high heterozygosity and allogamy of heterogenous and allogamous genotypes of *M. sativa* [29]. This result concords with those obtained by Heyn [46] who reported that all perennial species of *Medicago* genus are auto-tetraploid. Whereas it differs with that obtained by Abdelkefi et al. [47] who mentioned that *M. tunetana*. is diploid and each gene is presented by only two copies. All forage species are frequently allogamous which makes their genetic card study more complex and difficult to carry out [48]. This allogamy is one of the most important causes of genetic erosion as well as of initial allelic variation loss [49] and obtaining new characters. The observed allogamy in perennial *Medicago* species may be explained by the gametophytic self-incompatibility which makes getting a pure line so difficult or even impossible [48].

The molecular study for seven *M. tunetana* accessions and local variety of *M. sativa* using seven SSR markers had showed a large genetic diversity within *Medicago* genotypes. Two different groups of *Medicago* accessions were formed based on the generated dendrogram at 57% of similarity. Gardon et al. [50] showed high degree of variability among *M. sativa* genotypes with six SSR markers. Cholastova and Knotova [30] demonstrated that the estimation of genetic diversity of *Medicago* genotypes could be obtained even with only three SSR markers. Based on obtained results, used SSR markers were as efficient to determine genetic variation among the studied *M. tunetana* accessions. This work is the first genetic diversity study of *M. tunetana* genotypes; the number of SSRs used (Table 3) as well as the broad genetic base of *M. tunetana* could be considered not enough. Therefore, this work must be completed by other molecular study using more SSR markers in order to obtain more information about genetic variability of *M. tunetana* and karyotype analysis.

*Exsitu* conservation of *M. tunetana* accessions as a *Pastoritum* or seeds in Gene Bank contributes to this genetic resource conservation as well as in the local alfalfa breeding programs. *M. tunetana* is an allogamous plant species needing an optimal sample size more than 20 individuals in order to obtain reproducible bands to differentiate between accessions within an active collection. A population sample size lower than 20 individuals could increase the effect of genetic drift which generates loss of biodiversity. Sample size to be conserved in Gene Bank must be surrounding 200 seeds per accession which will allow preserving genetic diversity [51].

## 4. Material and Methods

### 4.1. Origin of Plant Material Collection 

In order to collect *M. tunetana* genotypes, many prospecting and collecting missions were carried out by visiting sites reported by Pottier-Alapetite [15] during five years (2015–2020) in the mountains of Western North and Western Centre regions of Tunisia that are located in three different Tunisian governorates (Siliana, Kef and Kasserine) (Table 1).These regions are characterized by a continental climate and generally roughly with an average annual rainfall ranging between 220 mm and 550 mm [4]. 

### 4.2. Sample Size

Six wild accessions of *M. tunetana* were collected for this work. For the morphological study, eight plants from each accession were randomly selected and used. For the genetic analysis, a local variety of *M. sativa* was added to be used as reference. Twenty plants per accession were randomly selected, and green healthy leaves from each plant were chosen for DNA extraction. This study was carried out in the molecular laboratory of National Gene Bank of Tunisia.

### 4.3. Ecological Analyses

The ecological characterization of each prospected site was determined in order to evaluate the most favourable environmental factors (climatic, soil and altitude) for *M. tunetana* genotypes development. In fact, the annual precipitation and number of rainy days for each prospected site were obtained for twenty-seven years (1992 to 2019) from the Ministry of Agriculture, Hydraulic Resources and Fisheries of Tunisia. Therefore, three representative soil samples from each site were taken and had been submitted to four physical analyses (clay%, fine silt %, coarse silt % and sand %) and five chemical analyses (pH, organic matter %, total and active limestone % and potassium content).

### 4.4. Morphological Traits

*M. tunetana* accessions variability was examined according to description given by Pottier-Alapetite [15] for its taxonomic classification. The morphological characterization was determined based on 48 samples and six *M. tunetana* genotypes. Eleven quantitative traits relating to leaves, pods and seeds were measured: leaflet length (LL), leaflet width (LW), petiole length (PL), petiolule length (PeL), stipule length (StL), pod length (PoL), pod width (PoW), number of pod turns (NPT), seed length (SL), seed width (SW) and seed thickness (ST). Data were measured using electronic digital calliper (0–150 mm) with an accuracy of 0.01 mm.

### 4.5. Molecular Analysis 

Genomic DNA was extracted from fresh leaf tissue plants of *M. tunetana* according to the cetyltri-methyl-ammonium bromide (CTAB) technique of [52] with slight modifications. Plant material was ground up in liquid nitrogen, resuspended in 1 mL of CTAB 2 × buffer and incubated at 60 °C for 30 min. After chloroform-isoamylic (24:1) extraction, the aqueous phase was collected and the nucleic acid was precipitated with cold isopropanol, then washed with 70% ethanol and resuspended in 1 × TE buffer after being air-dried. Genetic diversity of different accessions of *M. tunetana* was evaluated using seven SSR markers identifying QTL genes responsible of the plant height which are positively correlated with forage yield potential in the model species *M. truncatula* [37]. These microsatellites are saved in the databases of EST (expressed sequence tag) and selected by Julier et al. [37] as an efficient marker for molecular characterization of autotetraploid and allogamous plant’s *Medicago* genus (Table 6). The PCR was performed in a final volume of 25 µL containing 100 ng of genomic DNA, 0.5 µM of each primer pair, 2.5 µL of 1 × PCR buffer, 1.5 mM of MgCl_2_, 0.2 mM dNTPs and 0.2 µL of Taq DNA polymerase. DNA amplification was performed in a Bio Rad C1000 thermocycler programmed with 35 cycles: 1 cycle for initial Pre-Denaturation at 94 °C for 4 min; 35 cycles consist of denaturation at 94 °C for 1 min, Melting at Tm °C for 1 min and extension at 72 °C for 2 min;1 cycle for final extension at 72 °C for 3 min. PCR products (12 µL per sample) were separated in 3% agarose gel at 100 volts constant power for 90 min and then visualized with BET fluorescence in UV table.

### 4.6. Statistical Analysis

A two-way analysis of variance (ANOVA) was realized for soil analyses and quantitative morphological traits to evaluate difference between means which genotype was the first factor (MT1, MT2, MT3, MT4, MT5, MT6, MT7). Results were statistically significant at *p* < 0.05 for Tukey test using the statistical software SPSS statistics version 22.0 (IBM, Armonk, NY, USA). 

Morphological, ecological and edaphic traits were examined by a canonical correspondence analysis (CCA) in order to determine the relationship between the analysed traits and the similarity among accessions of *M. tunetana* using PAST software, version 4.03 [53]. The CCA was carried out to the data matrix (11 morphological traits, 6 ecological traits, 10 edaphic traits and 6 genotypes).

For the molecular traits, the bands visualized on the gel were identified to determine whether they were a single allele or just auto-amplification to be discarded. Faint and unreliable bands were not considered for the analysis. Amplified fragments were scored as 1 or 0 for bands presence or absence, respectively. Sizes amplification bands were estimated using DNA Leader 100 pb marker. Genetic distance between accessions was calculated according to Nei [42]. Unweighted pair group method using arithmetic average UPGMA dendrogram were drawn using the SAHN clustering of NTSYSpc 2.10, based on Genetic Distances.

## Figures and Tables

**Figure 1 plants-10-01923-f001:**
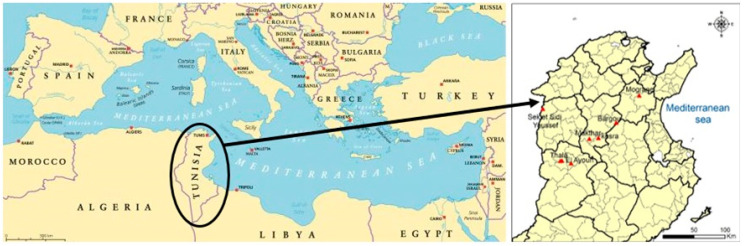
The geographical localization of the eight *M. tunetana* and *M. sativa* accessions [legend: Bargou: MT1; Makthar: MT2; Kesra: MT3; Sekiet Sidi Youssef: MT4; Thala: MT5–MT6; El Ayoun: MT7; Mograne: MS1].

**Figure 2 plants-10-01923-f002:**
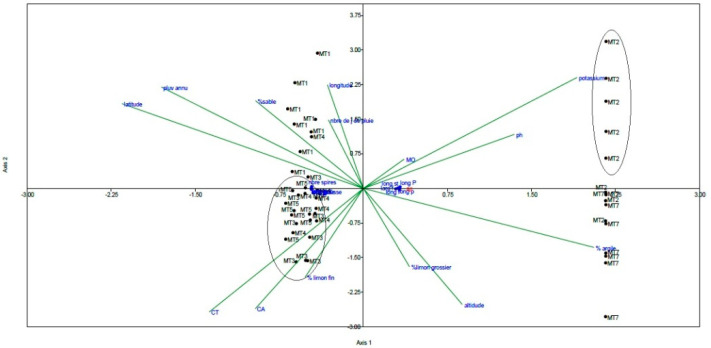
Canonical correspondence analysis (CCA) carried out with eleven morphological data and sixteen eco-geographical traits of seven accessions of *M. tunetana* collected from different locations of Tunisian ridge.

**Figure 3 plants-10-01923-f003:**
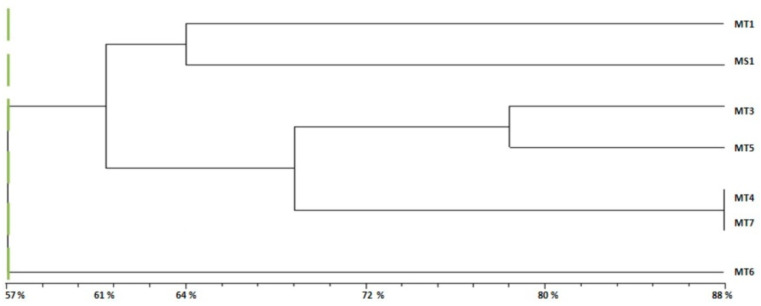
The UPGMA dendrogram based on the analysing of the molecular data by five SSR markers and showing similarity coefficients and genetic relationships among seven accessions of *M. tunetana* and variety of *M. sativa* used as a reference.

**Table 1 plants-10-01923-t001:** Eco-geographical variables for different studied accessions of *M. tunetana.* Means with the same letters are not significantly different (ANOVA, Tukey test at *p* < 0.05); nd: not determined.

N°	Eco-Geographical Factors	Accessions							
		MT1	MT2	MT3	MT4	MT5	MT6	MT7	MS1
1	Origin	Siliana, Bargou	Siliana, Makthar	Siliana, Kesra	Kef, Sekiet Sidi Youssef	Kasserine, Thala	Kasserine, Thala	Kasserine, El Ayoun	Zaghouan, Mograne
2	Latitude (dms)	36°2′47.15”	35°49′43.77”	35°49′43.774”	36°14′32.532”	35°30′39.39”	35°29′5.55”	35°20′50.42”	36°42′82.72”
3	Longitude (dms)	9°40′34.32”	9°21′28.38”	9°21′28.205”	8°22′8.291”	8°42′58.45”	8°44′30.34”	8°34′31.08”	10°09′20.49”
4	Altitude (m)	523	868	1010	899	1003	1041	903	156
5	Bioclimatic zone	SSACW	SSACW	SSACW	SHCW	MSACW	MSACW	MSACW	SSACW
6	Rainy days (In no)	63.44	58.50	61.84	59.52	46.08	84.46	54.83	85.68
7	Annual precipitation (mm)	522.50	468.73	494.89	464.80	340.83	327	299	415.7
8	Texture	Clay	Silty clay loam	Silty clay loam	Silty clay	Clay	Clay	Silty clay loam	nd
9	Fine silt (%)	19.85 ^b^	23.49 ^c^	23.32 ^c^	26.28 ^d^	20.05 ^b^	17.95 ^a^	26.06 ^d^	nd
10	Coarse silt (%)	2.55 ^a^	6.81 ^d^	6.15 ^c^	15.20 ^f^	5.30 ^b^	9.91 ^e^	9.96 ^e^	nd
11	Clay (%)	28.68 ^c^	37.2 ^de^	36.35 ^d^	25.16 ^b^	28.62 ^c^	21.87 ^a^	37.88 ^e^	nd
12	Sand (%)	42.21 ^e^	31.5 ^b^	33.29 ^c^	38.43 ^d^	42.74 ^f^	46.2 ^g^	27.12 ^a^	nd
13	K (ppm)	38 ^d^	102 ^ab^	297 ^c^	052 ^a^	137 ^b^	292 ^c^	247 ^c^	nd
14	Organic Matter (%)	4.91 ^c^	2 ^a^	8.6 ^e^	6.72 ^d^	2.41 ^a^	3.55 ^b^	4.32 ^bc^	nd
15	pH	7.47 ^ab^	7.2 ^a^	8.36 ^e^	7.92 ^cd^	8.1 ^de^	8.01 ^cd^	7.71 ^bc^	nd
16	Total limestone (%)	35.21 ^a^	50 ^cd^	34.4 ^a^	44.00 ^bc^	53.19 ^d^	39.42 ^ab^	49.68 ^cd^	nd
17	Active limestone (%)	8.46 ^a^	30.7 ^d^	11.74 ^ab^	18.51 ^bc^	21.96 ^c^	14.51^abc^	20.62 ^c^	nd

Bioclimatic zones were defined according to [38] coefficient. SSACW: Superior semi-arid at cool winter, SHCW: Sub-humid at cool winter, MSACW: Medium semi-arid.

**Table 2 plants-10-01923-t002:** Mean value of eleven morphological traits for six *M. tunetana* populations (mm). Means with the same letters are not significantly different (ANOVA, Tukey test at *p* < 0.05); nd: not determined.

N°	Traits	MT1	MT2	MT3	MT4	MT5	MT6	MT7
1	LL **	12.00 ^a^	nd	17.06 ^b^	14.94 ^ab^	11.62 ^a^	nd	nd
2	LW ***	4.50 ^a^	nd	6.75 ^b^	6.31 ^b^	3.31 ^a^	nd	nd
3	PL	8.00	nd	7.06	7.87	5.00	nd	nd
4	PeL	1.69	nd	1.87	1.69	1.25	nd	nd
5	StL	7.06	nd	7.31	7.31	5.69	nd	nd
6	PoL **	3.64 ^a^	nd	4.98 ^b^	3.51 ^a^	4.40 ^ab^	nd	nd
7	PoW ***	6.04 ^a^	nd	8.91 ^c^	6.81 ^b^	6.04 ^a^	nd	nd
8	NPT ***	4.24 ^b^	nd	3.33 ^k^	2.60 ^a^	2.70 ^a^	nd	nd
9	SL ***	2.73 ^b^	nd	3.01 ^c^	2.63 ^ab^	2.55 ^a^	nd	nd
10	SW	1.91	nd	2.07	2.08	1.99	nd	nd
11	ST ***	1.08 ^b^	nd	1.06 ^b^	0.84 ^a^	0.86 ^a^	nd	nd

LL: Leaflet Length, LW: Leaflet Width, PL: Petiole Length, PeL: Petiolule Length, StL: Stipule Length, PoL: Pod Length, PoW: Pod Width, NPT: Number of Pod Turns, SL: Seed Length, SW: Seed Width, ST: Seed Thickness. **: significant at the 99% confidence level; ***: significant at the 99.9% confidence level.

**Table 3 plants-10-01923-t003:** Characterization of five polymorphic microsatellites used in the study of the genetic diversity of *M. tunetana* and *M. sativa* accessions.

No.	Markers	No. of Alleles	No. of Polymorphic Bands	Mean NoBands/Locus	PIC
1	FMT13	7	5	1.4	0.21
2	MTIC338	12	8	1.5	0.38
3	MTIC343	19	6	3.16	0.49
4	MTIC82	6	4	1.5	0.26
5	B14B03	10	5	2	0.31
	Total	54	28	-	-
	Mean	10.8	5.6	1.19	0.33

PIC: Polymorphism Information Content.

**Table 4 plants-10-01923-t004:** Genetic variation statistics for all loci. * na = Observed number of alleles; * ne = Effective number of alleles [39]; * h = Nei’s [40] gene diversity; * I =Shannon’s Information index [41].

Accessions	na *	ne *	I *	h *
MT1	2.0000	1.6756	0.4032	0.5930
MT2	2.0000	1.6756	0.4032	0.5930
MT3	2.0000	1.2677	0.2112	0.3669
MT4	2.0000	1.9231	0.4800	0.6730
MT5	2.0000	1.7705	0.4352	0.6269
MT6	2.0000	1.9231	0.4800	0.6730
MT7	2.0000	1.5743	0.3648	0.5511
MS1	2.0000	1.2677	0.2112	0.3669
Mean	2.0000	1.6347	0.3736	0.5555
St. Dev	0.0000	0.2566	0.1076	0.1234

**Table 5 plants-10-01923-t005:** Nei’s original measurements of genetic identity (above diagonal) and genetic distance (below diagonal) [42].

Pop ID	MT1	MT2	MT3	MT4	MT5	MT6	MT7	MS1
MT1	----	0.7600	0.6000	0.5600	0.6400	0.6400	0.4800	0.6800
MT2	0.2744	----	0.6800	0.6400	0.8000	0.6400	0.7200	0.6800
MT3	0.5108	0.3857	----	0.4800	0.7200	0.6400	0.8800	0.7600
MT4	0.5798	0.4463	0.7340	----	0.6800	0.5200	0.5200	0.5600
MT5	0.4463	0.2231	0.3285	0.3857	----	0.6000	0.7600	0.6400
MT6	0.4463	0.4463	0.4463	0.6539	0.5108	----	0.6800	0.4800
MT7	0.7340	0.3285	0.1278	0.6539	0.2744	0.3857	----	0.6400
MS1	0.3857	0.3857	0.2744	0.5798	0.4463	0.7340	0.4463	----

**Table 6 plants-10-01923-t006:** SSR primers, their respective sequences, number of alleles and annealing temperature (Ta °C).

No.	Primers	Sequences 5′-3′	Repeated Motif	Ta (°C)
1	MTIC451	F: CGATCGGAACGAGGACTTTA	(AAG)6	52
		R: CCCCGTTTTTCTTCTCTCCT		
2	FMT13	F: GATGAGAAAATGAAAAGAAC	(GA)2GG(GA)9	52
		R: CAAAAACTCACTCTAACACAC		
3	MTIC338	F: TCCCCTTAAGCTTCACTCTTTTC	(CTT)5	56
		R: CATTGGTGGACGAGGTCTCT		
4	MTIC343	F: TCCGATCTTGCGTCCTAACT	(GAA)8	56
		R: CCATTGCGGTGGCTACTCT		
5	MTIC82	F: CACTTTCCACACTCAAACCA	(TC)11	55
		R: GAGAGGATTTCGGTGATGT		
6	MTIC432	F: TGGAATTTGGGATATAGGAA	(AG)6	55
		R: GGCCATAAGAACTTCCACTT		
7	B14B03	F: GCTTGTTCTTCTTCAAGCTC	(CA)9	55
		R: ACCTGACTTGTGTTTTATGC		

## Data Availability

Data are contained within the article.

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
