# Peer review of "Eco-Geographical, Morphological and Molecular Characterization of a Collection of the Perennial Endemic Species Medicago tunetana (Murb.) A.W. Hill (Fabaceae) from Tunisia"

_plants, 2021, doi:10.3390/plants10091923_

Round 1

Reviewer 1 Report

Dear Authors, please consider comments below. English is not my native language, but I feel that it can be corrected. Pay attention to the yellow marks in the attached file.

It is highly desirable to provide a map with the locations of the studied samples.

ABSTRACT

morphologicaltraits > morphological traits

Tunetana > tunetana

developped > developed

add more keywords

INTRODUCTION

[1] native pastures legumes are more than 960 where 336 are Mediterranean endemic species. >

[1] there are more than 960 species of pasture legumes, 336 of which are Mediterranean endemics.

But unfortunately, pasture genetic resources of in Tunisia still less developed yet despite it may serve to enrich Gene Bank and to breed many crop cultivars as well > But, unfortunately, the pasture genetic resources in Tunisia are still less developed, despite the fact that they can enrich the bank gene and include many cultural varieties in breeding.

environment-independent > environment independency

“Among these, DNA based markers; SSRs are accepted as the most reliable to estimate the level of variation and the relationships among alfalfa populations.” Delete.

“were identified in the EST database of M. truncatula was mapped also in M. sativa [35]” > “identified in the EST database of M. truncatula was mapped also in M. sativa [35”

“one population of M. sativa as a control” > “one population of M. sativa as a reference”

TABLE 1

  1. Tunetana (Murb.) Hill > M. tunetana

clay > Clay (in MT5 row)

Explain the meaning of abcdefg

2.2. MORPHOLOGIC VARIATION

LINE1: Morphologic > Morphological

TABLE 2.

Why MT7 is absent?

LINE16: “The letters represent the statistically difference between groups.” More explanation of symbols  is needed.

LINE35: delete ”presented”

LINES 35-36: “seven accessions of M. tunetana and a population of M. sativa” > “seven accessions of M. tunetana (MT1-7) and a population of M. sativa (MS1)

LINE 37: “markersused” > “markers used”

TABLE 3.

Delete “their number of alleles, their number of polymorphic bands and their mean value of the number of bands per locus”

Change first line of a table to:

No.

Marker

No. of allele

No of polymorphic band

Mean No Band/Locus

PIC

TABLE 4.

Locus > Accession

LINES 61-62 and further text and dedrogram: why MT7 accession is not considered?

LINE 62, 94, 161: control > reference

LINES 63-65 and further text and Figure 2:

The dendrogram is scaled in similarity coefficients, not differences. Since the text further refers to the percentage of similarities, the scale is also better to designate in percent.

I see on the dendrogram that the group 1 has not 67%, but about 57% of similarity with other samples, and MT6 with other samples is also similar to 57%

LINES 119-120: “one other group was formed by M. sativa population by cuting the dendrogram at a similarity coefficient of 64%” – there is no such group on the dendrogram.

LINE127: “considerate low as well as the broad genetic base of M. tunetana” – please revise this phrase

LINE 135:  “can increase the genetic drift” > “can increase the effect of genetic drift”

LINES 136-137: “The conservation of each plant sample which has got a 136 large genetic variability save the genetic selection” - please revise this phrase

  1. MATERIALS AND METHODS

Table 6 contains data repeated in Table1. Add Origin row and data for M. sativa in Table1 and exclude Table 6.

LINE 209: Staystical > Statistical

LINE 225: Un-weighted > Unweighted

Author Response

Dear Reviewer,

I would at first to thank you for all your corrections.

I took into consideration all your constructive comments and I corrected the manuscript according to it.

You find all corrections with a 'track changes' form in the revised manuscript.

Reviewer 2 Report

This manuscript reported the characterizations of a collection of the perennial endemic species Medicago tunetana in Tunisia in eco-geographical, morphological and molecular level. The experiments were well designed. The data are in general support their conclusions. I only have one suggestion. The quality of the figure 3 can be improved. The pictures are kind of blurred. 

Author Response

Dear Reviewer,

Thank you very much for your opinion about my manuscript

Concerning the figure 3, I delited it because of its quality and also it could not very expressive in the study. You find the correction (picture delete) in the revised version of the article with a 'track changes' form. please confirm it by accepting the correction on the word coorected version.